



# Dynamics of riverine $CO_2$ in the Yangtze River fluvial network and their implications for carbon evasion

Lishan Ran[1, 2], Xi Xi Lu[2, 3, *], Shaoda Liu[2]

[1]Department of geography, the University of Hong Kong, Pokfulam Road, Hong Kong
[2]Department of geography, National University of Singapore, 117570, Singapore
[3]College of Environment & Resources, Inner Mongolia University, Hohhot, 010021, China
[*]Correspondence: geoluxx@nus.edu.sg

**Abstract:** Understanding riverine carbon dynamics is critical for not only better estimates of various carbon fluxes but also evaluating their significance in the global carbon budget. As an important pathway of global land-ocean carbon exchange, the Yangtze River has received less attention regarding its vertical carbon evasion than lateral transport. Using long-term water chemistry data, we calculated $CO_2$ partial pressure ($pCO_2$) from pH and alkalinity and examined its spatial and temporal dynamics and the impacts of environmental settings. With alkalinity ranging from 415 to >3400 μmol $L^{-1}$, the river waters were supersaturated with dissolved $CO_2$, generally 2–20 folds the atmospheric equilibrium (i.e., 390 μatm). Changes of $pCO_2$ were collectively controlled by terrestrial ecosystems, hydrological regime, and rock weathering. High $pCO_2$ values were observed spatially in catchments with abundant carbonate presence and seasonally in the wet season when recent-fixed organic matter was exported into the river network. In-stream processing of organic matter facilitated $CO_2$ production and sustained the high $pCO_2$, although the alkalinity presented an apparent dilution effect with lower alkalinity concentrations in higher flow periods. The decreasing $pCO_2$ from the smallest headwater streams through tributaries to the mainstream illustrates the significance of direct terrestrial carbon input in controlling riverine carbon. With a basin-wide mean $pCO_2$ of 2662±1240 μatm, substantial



CO$_2$ evasion from the Yangtze River fluvial network is expected. Future research efforts are thus needed to quantify the amount of CO$_2$ evasion and assess its biogeochemical implications for watershed-scale carbon cycle. In view of the Yangtze River's relative importance in global carbon export, its CO$_2$ evasion would be significant for global carbon budget.

**Keywords:** CO$_2$ partial pressure ($p$CO$_2$); riverine carbon cycle; spatial and temporal patterns; CO$_2$ evasion; Yangtze River

## 1. Introduction

Inland waters, including rivers, streams, lakes, wetland, and reservoirs, have recently been

recognized as active components of the global carbon cycle, transporting, storing, and processing huge amounts of terrestrially-derived carbon (Aufdenkampe et al., 2011;Cole et al., 2007;Raymond et al., 2013;Richey et al., 2002;Weyhenmeyer et al., 2015;Borges et al., 2015). With a higher CO$_2$ partial pressure ($p$CO$_2$) than the atmospheric equilibrium (i.e., 390 µatm), inland waters are mostly net carbon sources to the atmosphere. Published studies show that the

annually degassed CO$_2$ from inland waters is estimated to almost entirely compensate the total annual carbon uptake by ocean systems (Wanninkhof et al., 2013;Regnier et al., 2013). Global estimates of CO$_2$ evasion from rivers and streams range from 0.56 to 1.8 PgC yr$^{-1}$ (Aufdenkampe et al., 2011;Raymond et al., 2013;Lauerwald et al., 2015). It is apparent that these results vary considerably and are associated with great uncertainties. The most recent estimate of 0.65 PgC

yr$^{-1}$ by Lauerwald et al. (2015) accounts for only 36% of the efflux estimate made by Raymond et al. (2013), although they used the same water chemistry data set. Among the numerous factors



contributing to current $CO_2$ evasion uncertainties, a principal reason is the absence of a spatially

explicit $pCO_2$ data set that covers the full spectrum of the global river and stream network.

Existing global maps of $CO_2$ evasion from fluvial network are typically generated on the basis of

incomplete spatial coverage of $pCO_2$, in which Asian rivers are heavily underrepresented (e.g.,

Aufdenkampe et al., 2011; Battin et al., 2009; Lauerwald et al., 2015; Raymond et al., 2013).

Due to lack of direct *in situ* measurements, simplified extrapolation is normally used to predict

$pCO_2$ in and $CO_2$ evasion from Asian river systems. Consequently, the estimation accuracy is

problematic and even erroneous. For example, for the Yellow River in East Asia, while the

calculated $pCO_2$ from river water chemistry records is 2800 µatm (Ran et al., 2015a), the

modeled $pCO_2$ by Lauerwald et al. (2015) is 30% lower (i.e., <2000 µatm). A much lower

estimate of <700 µatm can be derived from the $pCO_2$ map produced in Raymond et al. (2013).

Such great discrepancies are largely because riverine $pCO_2$ is highly site-specific and affected by

a wide range of environmental factors (e.g., Abril et al. 2015; Teodoru et a., 2015). Asian rivers

are significant contributors to global carbon export, accounting for 40% of the global carbon flux

from land to sea (Schlünz and Schneider, 2000;Hope et al., 1994). Estimating the amount of $CO_2$

degassed from Asian rivers is critical for global $CO_2$ evasion assessments. Recent work in

Mekong and Yellow rivers has demonstrated high $pCO_2$ and $CO_2$ effluxes (Alin et al., 2011;Ran

et al., 2015b), further highlighting the necessity of incorporating the currently underrepresented

Asian rivers into global carbon budget assessments.



As an important carbon contributor to the West Pacific Ocean, the Yangtze River has received widespread attention in fluvial carbon export at various spatial and temporal scales. Studies of

flux estimates of different carbon species date back to the early 1980s (Cauwet and Mackenzie, 1993;Gan et al., 1983;Milliman et al., 1984;Wang et al., 2012;Zhang et al., 2014;Ittekkot, 1988). Intensive observations covering seasonal variability show that the Yangtze River transports approximately 20 Mt of carbon per year into the oceans (Wu et al., 2007;Bao et al., 2015). Contrary to the long history of lateral export measurements, however, few studies have examined

the vertical carbon exchange between the river system and the atmosphere (Li et al., 2012;Zhao et al., 2013;Chen et al., 2008). This is by nature largely due to the differences in sampling strategy. Unlike the lateral export that only involves measurements on the mainstream or at specific sites near the river mouth, quantifying basin-wide $CO_2$ evasion requires a spatially explicit $pCO_2$ data set encompassing the entire fluvial network. Any attempts of using limited

local measurements to up-scale to the watershed scale are challenging and subject to large uncertainties. This has impacted the understanding of riverine carbon cycle within the Yangtze River watershed as well as its links to the atmosphere and ocean systems.

By using long-term water chemistry data measured in the Yangtze River basin, we calculated the

riverine $pCO_2$ from pH and alkalinity. In combination with hydrologic and geologic information, the objectives of this study were to 1) investigate the spatial and temporal patterns of $pCO_2$ under 'natural' processes before significant human perturbations, mainly dam impoundment and land use changes since the 1990s; 2) to explore the couplings between $pCO_2$ and environmental





settings by investigating environmental and geomorphologic controls. Based on the obtained

$pCO_2$, we further evaluated its implications for $CO_2$ evasion. In view of the Yangtze River's role

in global fluvial export of water, sediment, and carbon (Syvitski et al., 2005;Wang et al., 2012),

its $CO_2$ evasion would be globally substantial. This $pCO_2$ database is thus helpful to examine the

spatial distribution of global riverine $pCO_2$ and to refine global $CO_2$ evasion from river systems.

**2. Material and methods**

**2.1 The Yangtze River basin**

With a length of 6380 km, the Yangtze River is the longest river in China and the third longest in

the world. The river originates on the Tibetan Plateau and flows eastward through the Sichuan

Basin and the Middle-Lower Reach Plains, before emptying into the East China Sea (Fig. 1a). Its

drainage area is 1.81 million km$^2$. The Yangtze River basin is mainly overlain by sedimentary

rocks that are composed of marine carbonates, evaporites, and continental deposits. Carbonates

are widely distributed within the watershed and are particularly abundant in the Wujiang,

Yuanjiang, and Hanjiang tributary catchments (Fig. 1b). Silicates are also widely present in the

basin while metamorphic rocks are mainly scattered in the middle-lower reach (Fig. 1b). The

Yangtze River is joined by a number of large tributaries, including the Yalongjiang, Daduhe,

Minjiang, Jialingjiang, Wujiang, Yuanjiang, Xiangjiang, Hanjiang, and Ganjiang rivers (Fig. 1a).

**Figure 1**

Except the headwater region characterized by high elevation and cold climate (annual mean

temperature <4 °C), the remaining watershed is affected by a subtropical monsoon climate with



the annual mean temperature in the middle-lower reach varying from 16 to 18 ℃ (Chen et al.,

2002). Rainfall is the major source of water discharge, whereas snowfall supply is only

significant in the ice-covered upstream mountainous areas. With a mean precipitation of 1100

mm yr$^{-1}$, the precipitation is spatially highly variable, decreasing from 1644 mm yr$^{-1}$ in the

lower reach, to 1396 mm yr$^{-1}$ in the middle reach, and 435 mm yr$^{-1}$ in the upper reach (Chetelat

et al., 2008). Approximately 60% of the annual precipitation falls during the wet season from

June to September. Affected by summer monsoon, the wet season generally occurs earlier in the

middle and lower reaches than in the inland upper reach. The water discharge from the upper to

the lower reach presents a strong seasonal variability (Fig. 2). Monthly peak discharge occurs in

July and can be 5−7 times greater than the lowest discharge in the dry season. The mean

discharge at Datong station is 28,200 m$^3$ s$^{-1}$ (see its location in Fig. 3b), and consequently the

Yangtze River annually discharges 889 km$^3$ of water into the ocean (Yang et al., 2002).

**Figure 2**

**2.2 Water chemistry data**

Concentrations of alkalinity, major ions, and dissolved silica measured at 359 stations in the

Yangtze River watershed (Fig. 1a) during the period 1960s−1985 were retrieved from the

Hydrological Yearbooks, which were yearly produced by the Yangtze River Conservancy

Commission (YRCC) for internal use. Other environmental variables concurrently measured at

each sampling event, including pH, water temperature, and water discharge, were also extracted

from the yearbooks for this study. The water samples for pH and temperature measurement were

taken in the same period as these for ion analysis. The sampling frequency ranged from 1 to 14





times per month depending on flow conditions. Sampling at some stations during the period

1966−1975 was less frequent. About 80% of the 359 stations have been continuously sampled

for at least 10 years, starting from the early1970s. To avoid severe river pollution by human

activity, only the samples collected prior to 1985 were used. In addition, samples with the pH

lower than 6.5 were manually discarded because the calculated $p$CO$_2$ would be greatly biased

due to contributions of noncarbonated alkalinity such as organic acid anions (Abril et al.,

2015;Hunt et al., 2011). Because reservoir trapping and increased water residence time can

remarkably alter the physical and biogeochemical properties of running water (Kemenes et al.,

2011;Barros et al., 2011), the stations located inside or shortly below reservoirs were also

intentionally removed. Given the tidal influences, mainstream stations downstream of Datong,

626 km inland from the coast, were also excluded, as were the stations in the delta region that

were affected either by tides or by intersections with other rivers via artificial canals. Based on

these selection criteria, 339 stations, including 13 mainstream stations and 326 tributary stations,

were retained and 47,809 water chemistry measurements in total were compiled.


Chemical analyses of water samples were performed under the authority of YRCC following the

standard procedures and protocols described by Alekin et al. (1973) and the American Public

Health Association (1985). While pH and temperature were measured in the field, the alkalinity

was determined by acid titration. Detailed sampling and analysis procedures were presented in

Chen et al. (2002). One important issue regarding historical records is data reliability. No

assessment reports on quality assurance and quality control are available in the hydrological





yearbooks. An effective evaluation approach is to compare the hydro-chemical differences for

samples collected at the same station but by different agencies. The Wuhan station on the

Yangtze mainstream has also been monitored under the United Nations GEMS/Water

Programme since 1980 (only yearly means available at http://www.unep.org/gemswater). The

pH value from the yearbooks agreed well with that measured by the GEMS/Water Programme

with <1.8% differences, while the alkalinity discrepancy between the two data sets is larger

(Table 1). The yearbooks report a slightly higher alkalinity than the GEMS/Water Programme

results by 7.6–13.9%, indicating that the yearbook reports are reliable for $p$CO$_2$ calculation. High

data quality of the yearbook reports can also be validated from comparison of major dissolved

elements measured by the two agencies at Wuhan station (see Chen et al., 2002).

**Table 1**

### 2.3 Calculation of $p$CO$_2$

The conventional method of calculating $p$CO$_2$ from pH and alkalinity was used. With ~90% of

the measured pH ranging from 7.1 to 8.3 suggestive of natural process for the Yangtze River,

bicarbonates were assumed equivalent to alkalinity (Amiotte-Suchet et al., 2003) because 96% of

the alkalinity was dominated by bicarbonates. This approximation has been frequently used in

Chinese river systems (Yao et al., 2007;Li et al., 2012;Ran et al., 2015a). The $p$CO$_2$ was then

calculated using CO2SYS program (Lewis and Wallace, 1998). However, using this method

would produce biased extreme values that are unrealistic in natural river systems (Hunt et al.,

2011;Weyhenmeyer et al., 2015). We thus reported median values per sampling station instead

of means to avoid the impact of erroneous extreme results.



## 3. Results

### 3.1 Spatio-temporal variability of alkalinity and $p\mathrm{CO_2}$

Except the excluded measurements, pH in the Yangtze River waters varied from 6.5 to 9.2 with

96% of the pH measurements ranging from 7.3 to 8.3 (Table 2). Higher pH values (i.e., >7.8)

were spatially measured in the headwater streams and the Hanjiang catchments (see Fig. 1a for

location). In comparison, the tributaries in the southern part of the watershed exhibited relatively

lower pH values. For the mainstream (Table 2), the median pH showed a significant downstream

decrease from 8.29 to 7.55 ($r^2 = 0.77$; $p < 0.001$). The alkalinity varied from 415 to >3400 μmol

$L^{-1}$ (Fig. 3a). Higher alkalinity (i.e., >2500 μmol $L^{-1}$) was observed in the upper reach and the

upper part of the middle reach (Fig. 3a), in particular the carbonate-rich tributary catchments

(e.g., the Jialingjiang, Wujiang, and Hanjiang rivers). In contrast, the lower part of the middle

reach (mainly the Ganjiang River) and the lower reach showed a lower alkalinity of <2000 μmol

$L^{-1}$. The average alkalinity over the whole watershed was 2210±1023 μmol $L^{-1}$.

**Figure 3 and Table 2**

The calculated $p\mathrm{CO_2}$ varied by a magnitude of 2 with the highest $p\mathrm{CO_2}$ being 24,432 μatm. At 95%

of the stations, the $p\mathrm{CO_2}$ was higher than 1000 μatm, generally 2−20 folds the atmospheric $p\mathrm{CO_2}$.

Only one station in the upper reach showed a median $p\mathrm{CO_2}$ lower than the atmosphere. In the

mainstream, the $p\mathrm{CO_2}$ increased from ~700 μatm at the uppermost station to 3800 μatm at

Nanjing near the river mouth (Table 2). Averaged over all stations, the basin-wide $p\mathrm{CO_2}$ was

2662±1240 μatm. To better illustrate its spatial variability, we modeled the $p\mathrm{CO_2}$ for the whole



stream network using the Kriging interpolation method in ArcGIS 10.1 (Esri, USA) with the

assumption that the station-based $pCO_2$ was representative of the surrounding streams. Similar to

alkalinity, the $pCO_2$ presented significant spatial variations (Fig. 3b). The Yangtze mainstream

near the headwater region and the Yalongjiang catchment showed the lowest $pCO_2$, generally

<1000 µatm. In comparison, the carbonate-rich tributaries in the southern part of the watershed

had high $pCO_2$ values. With carbonates occupying 83% of the catchment, the Wujiang River

presented the highest median $pCO_2$ than other tributaries, averaging 3550±1356 µatm. In the

lower reach, the $pCO_2$ was 3988±1244 µatm on average, which is inconsistent with its relatively

low alkalinity of <2000 µmol $L^{-1}$ (Fig. 3a). It is worth noting that the $pCO_2$ in Hanjiang

catchment was lower than expected, given its high alkalinity (>2500 µmol $L^{-1}$). Differences in

pH in these catchments are likely a principal cause of these inconsistencies.


In addition, the $pCO_2$ also showed strong temporal variability. Fig. 4 presents an example of

$pCO_2$ changes at Datong station on the mainstream. Despite considerable inter-annual variations

that could change by a factor of 5, the annual $pCO_2$ declined steadily during the >20-year-long

sampling period ($r^2 = 0.18$; $p < 0.05$) (Fig. 4a). This trend is pronounced even if the anomalously

high values in the late 1960s are excluded from analysis. Indeed, more than half of the evaluated

stations, mainly in the middle-lower reach, showed a significant decreasing trend at the 95%

confidence level. In contrast, gradual increases were observed at some tributary stations in the

upper reaches. Seasonally, the $pCO_2$ in the wet season was on average 30% higher than that in

the dry season (Fig. 4b), and greater fluctuation ranges could be observed in the wet season.



**Figure 4**

### 3.2 Correlations with hydro-geochemical variables

Fig. 5 presents two representative examples showing responses of alkalinity and $pCO_2$ to

hydrological regimes. Changes of alkalinity at both stations reflected a clear dilution effect. High

alkalinity concentrations were measured in low flow periods when groundwater was the major

contributor to runoff (Figs. 5a and 5c). Checking all stations indicated that the alkalinity at 98%

of the stations decreased exponentially with increasing water discharge after the onset of the wet

season. In contrast, the $pCO_2$ presented diverse relationships with water changes (Figs. 5b and

5d).There was no discernible dependence of $pCO_2$ on flow in the mainstream, while a positive

correlation was widely observed in small tributaries. Although only two stations were plotted

here, these diverse responses of alkalinity and $pCO_2$ to flow changes were widespread within the

watershed, in particular for $pCO_2$ between mainstream and small tributaries.

**Figure 5**

In order to elucidate the impacts of rock weathering on $pCO_2$, we selected three typical tributary

catchments with differing rock compositions (Table 3). The Wujiang catchment is mainly

underlain by carbonates (83%) and the Ganjiang catchment by silicates (65%), whereas the

Jialingjiang catchment lies in the middle regarding the areal coverage of the two rocks (Table 3

and Fig. 1b). As the most typical weathering products of carbonate and silicate rocks, we plotted

$Ca^{2+}$ and dissolved silica (expressed as $SiO_2$) against $pCO_2$, respectively (Fig. 6). For the three

catchments with contrasting rock compositions, the $pCO_2$ showed different responses to $Ca^{2+}$ and

$SiO_2$. In Wujiang catchment, the log-transformed $pCO_2$ (i.e., lg($pCO_2$)) presented a significant



negative correlation with $Ca^{2+}$ concentration ($p < 0.001$) (Fig. 6). This negative correlation

became less apparent with decreasing carbonate coverage in Jialingliang and Ganjiang

catchments. In contrast, while the $lg(pCO_2)$ showed positive correlation with $SiO_2$ in Jialingjiang

and Ganjiang catchments characterized by high silicate coverage, no clear relation between

$lg(pCO_2)$ and $SiO_2$ was detected in Wujiang catchment (Fig. 6).

**Figure 6 and Table 3**

## 4. Discussion

### 4.1 Uncertainty analysis of $pCO_2$

As an important parameter for $CO_2$ evasion estimation, an accurate riverine $pCO_2$ is essential to

quantify $CO_2$ evasion and explore its biogeochemical implications for carbon cycle at different

scales. Compared with direct measurement by means of membrane equilibration or headspace

technique, the conventional $pCO_2$ calculation from alkalinity has been criticized for causing

biases (Long et al., 2015;Hunt et al., 2011). Huge overestimations (i.e., >100%) have been

reported in rivers with organic-rich and acidic waters due to combined effects of high organic

acids and low buffering capacity of carbonate systems at low pH (Abril et al., 2015).

Unfortunately, there were no organic carbon information in the yearbooks, and measurements of

dissolved organic carbon (DOC) in the Yangtze River started in the early 1980s. Its DOC

ranging from 1.3 to 1.5 mg $L^{-1}$ was relatively low compared with other major world rivers (Bao

et al., 2015;Wang et al., 2012). Our recent sampling also shows that the mean DOC is 1.9 mg $L^{-1}$

for the mainstream and 2.4 mg $L^{-1}$ for major tributaries (Liu et al., 2016). Given the neutral to

basic pH range and the alkalinity variations, we believe the impact of organic acids is minimal,



although a slight overestimation may have occurred as suggested by Abril et al. (2015). Our

recent $p$CO$_2$ measurements in the mainstream and major tributaries using a membrane contactor

(Qubit DCO$_2$ System, Qubit Biology Inc., Canada) also indicate that the calculated $p$CO$_2$ results

are consistent with the measured values with only ~8% differences (Liu et al., 2016).

Furthermore, this $p$CO$_2$ calculation method is sensitive to pH changes. High accuracy of pH

measurements is critical to reduce the associated uncertainty. Similar to other water chemistry

records (i.e., Butman and Raymond, 2011; Lauerwald et al., 2015; Weyhenmeyer et al., 2015),

the retrieved pH was reported with a precision of one decimal place. If the uncertainties in pH

measurement accuracy are assumed to 0.1 pH units, the calculated $p$CO$_2$ would be

underestimated by 26% or overestimated by 21%. To minimize human-induced disturbances in

the chemical equilibrium of natural waters, we excluded the samples with pH<6.5 and treated

them as being significantly polluted. Taking into account the higher alkalinity than the

GEMS/Water Programme results, the propagated uncertainty ranges from 14% (underestimation)

to 27% (overestimation). The Yangtze River basin is China's major industrial and agricultural

regions, and influences of human activity, such as sewage inputs and chemical fertilizer usage to

a lesser extent, may have altered the river water's chemical compositions and pH. As a result,

498 measurements were discarded from the analysis. Overall, for the Yangtze River with a high

buffering capacity of carbonate alkalinity and low DOC concentrations, the calculated $p$CO$_2$ is

reasonable and can be used for further CO$_2$ evasion estimation.



### 4.2 Environmental impacts on alkalinity and $p$CO$_2$

Export of alkalinity in river systems was affected by hydrological regime with a clear dilution

effect (Fig. 5). The average alkalinity was 35% lower in the wet season than in the dry season. In

both the mainstream and the tributaries, the higher alkalinity during low flow periods in the dry

season (Figs. 5a and 5c) illustrated the contribution of groundwater recharge in providing

abundant alkalinity. With widespread carbonate presence, groundwater in the Yangtze River

watershed was rich in dissolved inorganic carbon (DIC). Recent studies show that the alkalinity

of typical karst groundwater in the watershed is in the range of 3300–4200 $\mu$mol L$^{-1}$ (Li et al.,

2010b;Li et al., 2010a). With reduced relative contribution of groundwater in the wet season, the

high alkalinity was diluted by local rain events that carried lower DIC contents. Spatially, the

dilution effect was more pronounced in the upper reach than the middle-lower reach. This may

have revealed the response of alkalinity production to land cover. Catchments with a higher

forest cover normally exhibit a stronger dilution effect than cropland catchments (Raymond and

Cole, 2003). While cropland was the major land use type in the middle-lower reach accounting

for 53.5% of the total catchment area, forest cover in the upper Yangtze River watershed was

much higher (37.3%) than the middle-lower reach (30.4%; data are from Data Center for

Resources and Environmental Sciences for the 1980s).


Riverine dissolved CO$_2$ originates primarily from terrestrial ecosystem respiration, groundwater

input, and in-stream processing of land-derived organic matter (Wallin et al., 2013;Lynch et al.,

2010). Different from alkalinity showing a clear dilution effect, the stable $p$CO$_2$ in the Yangtze



mainstream likely reflected the impacts of different biogeochemical processes in maintaining its

$pCO_2$ (Fig. 5b). Compared to the dry season in which the $pCO_2$ was mainly controlled by DIC

inputs from groundwater, the elevated $pCO_2$ in the wet season suggested the influence of organic

carbon transport and decomposition. Its organic carbon content in the wet season is higher and

the age is younger due primarily to strong erosion and leaching of recent-fixed organic matter

(Wang et al., 2012;Zhang et al., 2014). Rapid mineralization of the labile fraction of organic

carbon can increase the $pCO_2$. For instance, approximately 60% of the recent-fixed carbon

entering the Yangtze River can be quickly degraded in the wet season, while the degradation

ratio in the dry season is only 31% (Wang et al., 2012). On the other hand, the increasing $pCO_2$

with flow in tributaries (Fig. 5d) indicated enhanced supply of fresh dissolved $CO_2$ during high

flow periods. For tributaries with more homogeneous catchment environments, decomposition of

soil organic matter can provide abundant dissolved $CO_2$ (Liu et al., 2016;Li et al., 2012), thus

generating a positive $pCO_2$ response to flow changes. From this perspective, the stable $pCO_2$ in

the mainstream implied that the enhanced dissolved $CO_2$ input by soil organic matter

decomposition from one region has likely been counteracted by low $pCO_2$ waters derived from

other regions. Consequently, the $pCO_2$ dynamics appeared to be independent of hydrograph. This

is highly possible given the spatial heterogeneity of the watershed environment in terms of

vegetation cover, soil type, and rainfall intensity.

The spatial distribution of alkalinity overlapped well with the carbonate outcrops (Figs. 1b and

3a), with ~60% of the high alkalinity concentrations measured in carbonate catchments. Using



$Ca^{2+}$ as a proxy of rock weathering, the strong correlation between $Ca^{2+}$ and alkalinity suggested

the dominant role of weathering in controlling alkalinity and DIC export (Fig. 7). This is

consistent with the significant impact of weathering on alkalinity as observed in other rivers

(Raymond and Cole, 2003;Humborg et al., 2010). Particularly, given the higher susceptibility of

carbonates to weathering than silicates (Goudie and Viles, 2012), the abundant carbonate

presence in Wujiang catchment helped to sustain its high alkalinity and $pCO_2$ (Table 3). The

negative response of $pCO_2$ to $Ca^{2+}$ in Fig. 6 indicated that an elevated pH has probably occurred,

offsetting the weathering-induced DIC inputs in affecting $pCO_2$. A slight pH increase would

result in a reduced $pCO_2$ as this calculation method is sensitive to pH fluctuations (Laruelle et al.,

2013). The positive correlation between $pCO_2$ and $SiO_2$ in Jialingjiang and Ganjiang catchments

demonstrated the impact of DIC export by silicate weathering. Despite the high silicate

weathering rate in Ganjiang catchment, its alkalinity represented only one third of that in the

other two catchments (Table 3). Apparently, its high $pCO_2$ of 2642±626 μatm was primarily due

to its low pH (~6% lower). Overall, the catchments with more carbonate presence presented

higher $pCO_2$ values (Figs. 1 and 3b).

**Figure 7**

Because $pCO_2$ was calculated from alkalinity, its spatial variability reflected largely the export of

the latter. The inconsistencies between $pCO_2$ and alkalinity in Hanjiang catchment were likely

caused by dam operation (Fig. 3). By altering the physical and biogeochemical properties of

flowing water, dam trapping could cause a greatly declined $pCO_2$ as a result of photosynthetic

$CO_2$ fixation and increased pH (Ran et al., 2015a). The Danjiangkou Reservoir (storage: 17.5



$km^3$) on the upper Hanjiang River was constructed in 1968. Unfortunately, the retrieved data for the Hanjiang River started from the 1970s, rendering it impossible to compare the $pCO_2$ differences between pre- and post-dam periods. An indirect evidence is that an elevated pH within the reservoir has been measured (7.95−8.33; Li et al., 2009) relative to the 1970s

(7.84±0.15). In the lower reach near the estuary (Fig. 3b), heterotrophic ecosystems and human activity could explain its high $pCO_2$. Settling down of particulate organic matter coupled with nutrient-rich water plume from offshore can accelerate $CO_2$ production. Chen et al. (2008) concluded that aerobic respiration of heterotrophic ecosystems was the primary determinant of the high $pCO_2$ in the inner Yangtze estuary. Moreover, the lower Yangtze River basin was highly

populated. Inputs of acids from agricultural fertilizer, sewage, and acid deposition have also decreased pH and shifted the carbonate system towards $CO_2$ (Duan et al., 2007;Chen et al., 2002), generating high $pCO_2$ values regardless of its relatively low alkalinity.

**4.3 Geomorphological controls on alkalinity and $pCO_2$**

To illustrate the geomorphological controls, the used 339 stations were aggregated by stream order based on their spatial positions. Both alkalinity and $pCO_2$ showed a decreasing trend from the smallest headwater streams through tributaries to the Yangtze mainstream (Fig. 8). The average decrease of alkalinity and $pCO_2$ were 94 $\mu mol\ L^{-1}$ and 266 $\mu atm$, respectively. Higher alkalinity and $pCO_2$ in the headwater streams reveal the significance of direct terrestrial inputs of

organic carbon and dissolved $CO_2$ in controlling riverine carbon cycle. Over the study period, the Yangtze River watershed suffered severe soil erosion, averaging 2167 $km^{-2}\ yr^{-1}$ (Wang et al.,



2007b). Huge amounts of carbon were transported into the river system via erosion (Wu et al.,

2007). Consequently, soil respiration and decomposition of the terrestrial-origin organic carbon

have resulted in the $CO_2$ excess in the headwater streams (Li et al., 2012).

**Figure 8**

The decreasing $pCO_2$ with increasing stream order imply continued $CO_2$ evasion along the river

continuum and reduced supply of fresh $CO_2$. Except the three lakes connected to the mainstream

(Fig. 1a), the Yangtze River network is largely confined to its channel. Without large floodplains

supplying labile organic matter to sustain high $pCO_2$ as in the Amazon River (Mayorga et al.,

2005), its $pCO_2$ decreased progressively from the headwaters towards the mainstream. In

addition, it is interesting to note that the $pCO_2$ in the highest three orders was equivalent (~1800

μatm; Fig. 8). Instead of continuous decline, the stable $pCO_2$ suggests a balance between $CO_2$

evasion and supply of fresh $CO_2$ from upstream catchments or aquatic respiration. Contrary to

the headwater streams with close contact with terrestrial ecosystems, the downstream large

streams and rivers are far away from rapid fresh $CO_2$ input. Moreover, these large streams and

rivers are generally characterized by low gas transfer velocity owing to weakened turbulence and

mixing with benthic substrates (Butman and Raymond, 2011;Borges et al., 2015). This can

effectively inhibit $CO_2$ degassing and therefore maintain the balance.

It is important to note, however, that the delineated 8 stream orders may not necessarily represent

the actual stream network. Limited by spatial resolution, the smallest headwater streams might

have been missed from the identified river network. In addition, these headwater streams are also





generally absent of sampling stations. With much closer biogeochemical interactions with land

ecosystems, these missed headwater streams tend to have higher $pCO_2$ (Benstead and Leigh,

2012;Aufdenkampe et al., 2011). Thus, the actual $pCO_2$ gradient along the stream order may be

sharper if a higher $pCO_2$ in the headwater streams is included.

### 4.4 Implications for riverine CO$_2$ evasion

As mentioned earlier, riverine carbon transport has been a significant component of carbon cycle.

Quantifying riverine carbon export is essential to better evaluate global carbon budget and

elucidate the magnitude of carbon exchange between different carbon pools. For the estimation

of $CO_2$ evasion, riverine $pCO_2$ denotes $CO_2$ concentration gradient across the water-air interface

and thus the potential of $CO_2$ exchange. Prior studies indicate that elevated riverine $pCO_2$ can

enhance $CO_2$ evasion due to a steeper concentration gradient and a greater $CO_2$ availability for

degassing (Long et al., 2015;Billett and Moore, 2008). When assessing global-scale $CO_2$ evasion,

however, the spatial distribution of $pCO_2$ is heavily skewed towards Northern America, Europe,

and Australia (e.g., Lauerwald et al., 2015; Raymond et al., 2013), while data for Asian rivers are

extremely lacking. This absence of an equally distributed $pCO_2$ database has made it challenging

to accurately estimate global $CO_2$ evasion. The role of Asian rivers in global carbon export

explicitly demonstrates that under-representation of Asian rivers would cause huge biases.

Comparing the Yangtze River with other rivers shows that its $pCO_2$ is higher than most world

rivers (Table 4). The average $pCO_2$ of 2662 µatm suggests that the Yangtze River waters are



potentially a prominent carbon source for the atmosphere. Large $CO_2$ evasion fluxes have been

reported by several small-scale studies in the upper reach and the estuary (Zhai et al., 2007;Chen

et al., 2008;Li et al., 2012), as also shown in Table 4. Nonetheless, a systematic estimation of

$CO_2$ evasion from the whole Yangtze River network, including mainstream and its tributaries of

all orders, remains lacking. This has further hampered the assessment of its $CO_2$ evasion in a

wider context linking the watershed's land-atmosphere and land-ocean carbon exchanges.

**Table 4**

Accelerated human activity is another urgent issue to be considered when investigating its

riverine $p$CO$_2$ and $CO_2$ evasion. Approximately 50,000 dams, including the world's largest

reservoir (i.e., the Three Gorges Reservoir; TGR), have been constructed in recent decades (Xu

and Milliman, 2009). Assessing the impacts of dam-triggered changes to flow regime and

biogeochemical processes on $p$CO$_2$ and $CO_2$ evasion is particularly important for deeper insights

into its riverine carbon cycle (Table 4). For example, while the $p$CO$_2$ at Datong station declined

continuously before the TGR impoundment (Fig. 4a; Wang et al., 2007a), our recent field survey

shows that it has recovered from 1440 µatm in the 1980s to present 1700 µatm (see Fig. 4a). As

for $CO_2$ degassing, recent work in the TGR indicates that its $CO_2$ evasion fluxes are different

from natural rivers  and are higher than other temperate reservoirs (Table 4; Zhao et al., 2013).

Future research efforts are warranted to conduct systematic monitoring and evasion estimation.

Given the Yangtze River's role in global carbon export, a comprehensive assessment of $CO_2$

evasion is also meaningful for global carbon budget.



**Conclusions**

The degassing of $CO_2$ from inland waters was recently incorporated into assessments of the global carbon budget. Because direct degassing measurements are time-consuming and unrealistic at large spatial scales, $CO_2$ efflux from aquatic systems is generally estimated from the water-air gradient of $pCO_2$ and gas transfer velocity. As an important parameter for calculating $CO_2$ evasion, riverine $pCO_2$ is affected by a variety of factors and could vary substantially over space and time. Based on water chemistry data measured in the Yangtze River basin during the period 1960s–1985, we calculated the $pCO_2$ from pH and alkalinity. The pH in the Yangtze River waters varied from 6.5 to 9.2 and the alkalinity ranged from 415 to >3400 $\mu$mol $L^{-1}$ with high alkalinity concentrations occurring in carbonate-rich tributary catchments. Except one station in the upper reach showing a lower $pCO_2$ than the atmosphere, the Yangtze River waters were supersaturated with dissolved $CO_2$, generally 2–20 folds the atmospheric equilibrium. Averaged over all stations, the basin-wide $pCO_2$ was 2662±1240 µatm.

The observed spatial and temporal variations of $pCO_2$ were collectively controlled by terrestrial ecosystems, hydrological regime, and rock weathering. High $pCO_2$ values were observed spatially in catchments with abundant carbonate presence and seasonally in the wet season when recent-fixed organic matter was flushed into the river network. Decomposition of organic matter by microbial activity in aquatic systems facilitated $CO_2$ production and sustained the high $pCO_2$ values in the wet season, although the alkalinity presented a significant dilution effect with water discharge. In addition, the $pCO_2$ decreased with increasing stream orders from the smallest




headwater streams through tributaries to the mainstream. A higher $p$CO$_2$ in the headwater streams illustrated the influence of direct inputs of terrestrially-derived organic matter and weathering products via erosion and flushing on riverine carbon dynamics.

The substantially higher $p$CO$_2$ than the atmosphere indicated a potential of significant CO$_2$ evasion from the Yangtze River fluvial network. Estimating the amount of CO$_2$ evasion should be a top priority, upon which its biogeochemical implications for watershed-scale carbon cycle can be assessed in association with carbon burial and downstream export. Given the significant anthropogenic perturbations in recent decades, special attention must be paid to the resulting

changes to riverine $p$CO$_2$ and CO$_2$ evasion. Considering the Yangtze River's relevance to global carbon export, quantifying its CO$_2$ evasion is of paramount importance for better assessments of global carbon budget.

**Acknowledgements:** This work was financially supported by the National University of
Singapore (grants R-109-000-172-646 and R-109-000-191-646).

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



Table 1. Comparison of alkalinity (meq L$^{-1}$) and pH at Wuhan station between the GEMS/Water
Programme results and the hydrological yearbooks, expressed as mean±standard error.

| Item | 1980 | 1981 | 1982 | 1983 | 1984 | 1984 |
|---|---|---|---|---|---|---|
| | | | GEMS/Water Programme | | | |
| Alkalinity | 2.05±0.29 | 2.00±0.19 | 2.01±0.23 | 1.84±0.25 | 2.20±0.25 | 1.99±0.22 |
| pH | 7.83±0.16 | 7.73±0.24 | 8.04±0.09 | 8.06±0.05 | 8.00±0.09 | 7.88±0.06 |
| | | | Hydrological yearbooks | | | |
| Alkalinity | 2.31±0.31 | 2.19±0.24 | 2.27±0.27 | 2.03±0.30 | 2.38±0.28 | 2.31±0.24 |
| pH | 7.93±0.09 | 7.87±09 | 8.01±0.09 | 7.94±0.08 | 7.93±0.10 | 7.98±0.08 |





Table 2. Riverine pH, alkalinity, and $pCO_2$ in the Yangtze River basin (median±standard deviation).

| River/tributary | Station | pH | Alkalinity μmol L$^{-1}$ | $pCO_2$ μatm |
|---|---|---|---|---|
| Mainstream | Benzilan | 8.29±0.11 | 2352±435 | 681±156 |
| | Shigu | 8.18±0.48 | 2544±438 | 846±262 |
| | Jingjiangjie | 8.11±0.12 | 2905±362 | 916±202 |
| | Dukou | 8.22±0.12 | 2399±429 | 826±197 |
| | Longjie | 8.23±0.17 | 2185±396 | 786±226 |
| | Huatan | 8.17±0.15 | 2237±418 | 882±287 |
| | Pingshan | 8.13±0.10 | 2215±407 | 1001±235 |
| | Zhutuo | 7.88±0.19 | 2299±349 | 2405±781 |
| | Cuntan | 8.08±0.11 | 2173±311 | 1087±319 |
| | Yichang | 7.95±0.15 | 2343±300 | 1653±469 |
| | Luoshan | 7.76±0.11 | 2280±248 | 2380±691 |
| | Wuhan | 7.93±0.11 | 2060±263 | 1521±497 |
| | Datong | 7.84±0.14 | 1919±312 | 1711±806 |
| | Nanjing[a] | 7.56±0.16 | 2339±339 | 3796±1623 |
| | Nanjing[b] | 7.54±0.18 | 2296±357 | 3793±2186 |
| Major tributaries[c] | | | | |
| Yalongjiang | Xiaodeshi | 8.02±0.22 | 2576±465 | 1567±715 |
| Daduhe | Fuluzhen | 7.66±0.23 | 1909±289 | 2577±1620 |
| Minjiang | Gaochang | 8.02±0.15 | 1816±327 | 1020±525 |
| Tuojiang | Lijiawan | 8.01±0.11 | 2705±507 | 1504±572 |
| Jialingjiang | Beibei | 8.11±0.14 | 2289±509 | 1196±244 |
| Wujiang | Wulong | 8.01±0.14 | 2420±279 | 1361±508 |
| Yuanjiang | Taoyuan | 7.61±0.25 | 1822±480 | 2801±2144 |
| Xiangjiang | Xiangtan | 7.76±0.44 | 1739±331 | 2349±2521 |
| Hanjiang | Xiaoshicun | 7.93±0.13 | 2262±480 | 1715±536 |
| Ganjiang | Waizhou | 7.44±0.44 | 880±236 | 2205±2048 |
| Yangtze basin[d] | 1% percentile | 7.03 | 556 | 788 |
| | 10% percentile | 7.35 | 842 | 1236 |
| | 50% percentile | 7.71 | 2237 | 2455 |
| | 90% percentile | 8.05 | 3305 | 4344 |
| | 99% percentile | 8.28 | 4437 | 6163 |

[a] affected by high tides.

[b] affected by low tides.

[c] Median values of the data for the lowermost station on the mainstream of the specific tributary.



[d]Statistics based on the measurements at the used 339 stations.


Table 3. Hydro-geochemical features of the Wujiang (Wulong station), Jialingjiang (Wusheng station), and Ganjiang (Xiajiang station) catchments.

| Control station | Control area | Water discharge | pH | Alkalinity | $pCO_2$ | $Ca^{2+}$ | $SiO_2$ | Rock types (% of area) | | |
|---|---|---|---|---|---|---|---|---|---|---|
| | $km^2$ | $m^3 s^{-1}$ | | $\mu mol\ L^{-1}$ | $\mu atm$ | $\mu mol\ L^{-1}$ | $\mu mol\ L^{-1}$ | Carbonate | Silicate | Igneous + metamorphic |
| Wulong | 80,536 | 1570 | 7.72±0.14 | 3021±527 | 3537±1247 | 1145±278 | 59±31 | 82.9 | 14.8 | 2.3 |
| Wusheng | 80,550 | 793 | 7.80±0.21 | 2484±948 | 2671±490 | 1005±170 | 94±30 | 30.4 | 55.3 | 14.3 |
| Xiajiang | 62,387 | 1644 | 7.34±0.08 | 953±266 | 2642±626 | 242±91 | 105±18 | 9.1 | 64.7 | 26.2 |


Table 4. Comparison of $pCO_2$ and $CO_2$ evasion among world large rivers and typical reservoirs in the Yangtze River basin.

| River | Country | Climate | $pCO_2$ | $CO_2$ evasion | Reference |
|---|---|---|---|---|---|
| | | | $\mu atm$ | $mol\ m^{-2}\ yr^{-1}$ | |
| Yangtze network | China | Subtropical monsoon | 2662±1240 | / | This study |
| Upper Yangtze | China | Subtropical monsoon | 2100 | 57 | Li et al., 2012 |
| Lower Yangtze | China | Subtropical monsoon | 1297±901 | 14.2–54.4 | Wang et al., 2007a |
| Yangtze estuary | China | Subtropical monsoon | 650–1440 | 15.5–34.2 | Zhai et al., 2007 |
| Amazon | Brazil | Tropical | 3929 | 162.2 | Lauerwald et al. 2015 |
| Ottawa | Canada | Temperate | 1200 | 14.2 | Telmer and Veizer, 1999 |
| Hudson | USA | Temperate | 1125±403 | 5.8–13.5 | Raymond et al., 1997 |
| York estuary | USA | Temperate | 1070±867 | 6.3 | Raymond et al., 2000 |
| Mississippi | USA | Temperate | 1335±130 | 98.5±32.5 | Dubois et al., 2010 |
| Yukon | Canada | Subarctic | 582–705 | 11.6–21.2 | Lauerwald et al., 2015 |
| Yellow | China | Arid and semiarid | 2810±1985 | 312.4±149.2 | Ran et al., 2015b |
| Xijiang (Pearl) | China | Subtropical monsoon | 2600 | 69.2–130 | Yao et al., 2007 |
| Mekong (>100 m wide rivers) | SE Asia | Tropical monsoon | 703–1597 | 32–138 | Alin et al., 2011 |
| Godavari estuary | India | Tropical monsoon | <500–33,000 | 52.6 | Sarma et al., 2011 |
| Global rivers | | | 2400 | 131.2 | Lauerwald et al. 2015 |
| *Typical reservoirs in the Yangtze River basin* | | | | | |
| Wujiang cascade reservoirs | | | 38–3300 | -3.3–32.5 | Wang et al., 2011 |
| Three Gorges Reservoir (TGR) | | | / | 35.1 | Zhao et al., 2013 |



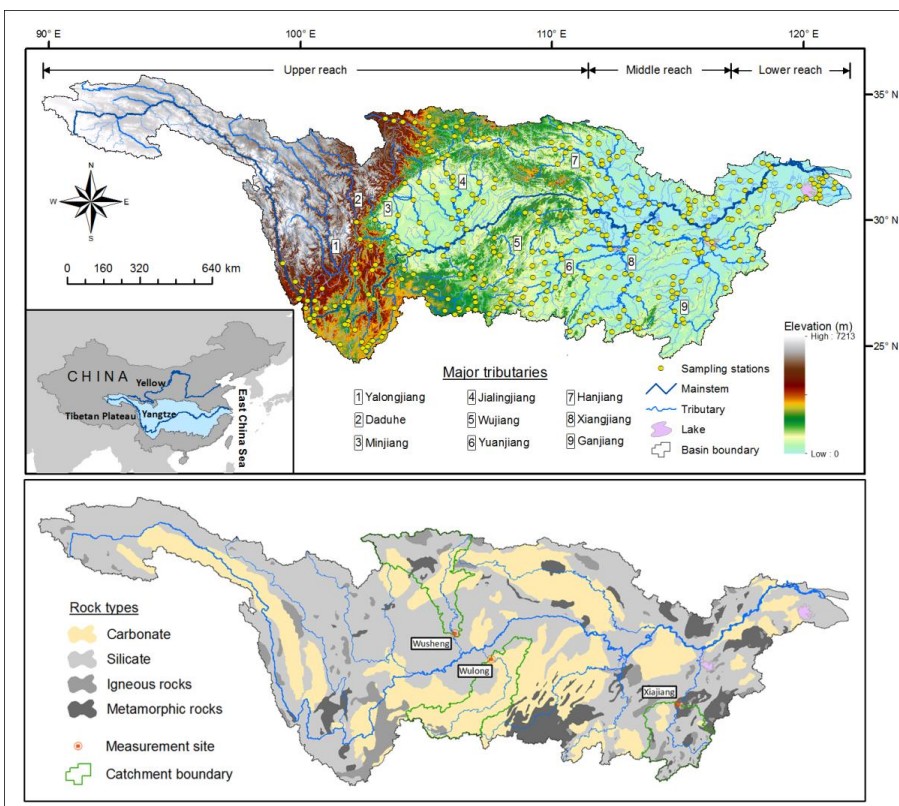

Fig. 1. Maps of the Yangtze River basin showing sampling stations (top) and rock compositions
(bottom). Rock information is modified from Chen et al. (2002) and Chetelat et al. (2008).




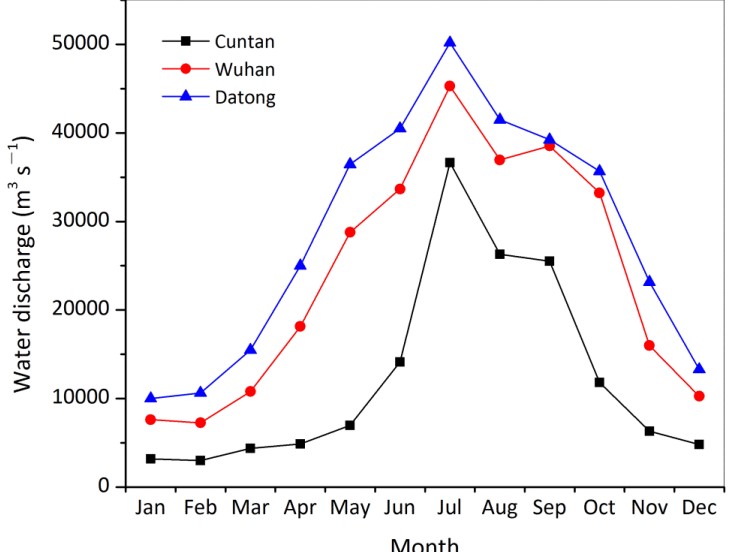

Fig. 2. Monthly variations in water discharge of the Yangtze River at Cuntan (upper reach), Wuhan (middle reach), and Datong stations (lower reach).


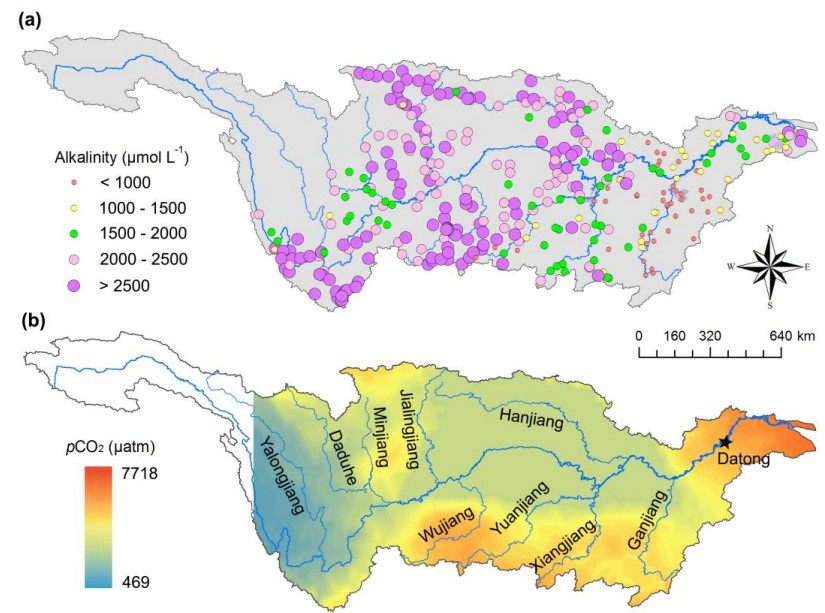

Fig. 3. Spatial distribution of alkalinity (a) and $p$CO$_2$ (b) in the Yangtze River basin. The headwater region in (b) was not interpolated due to insufficient stations.

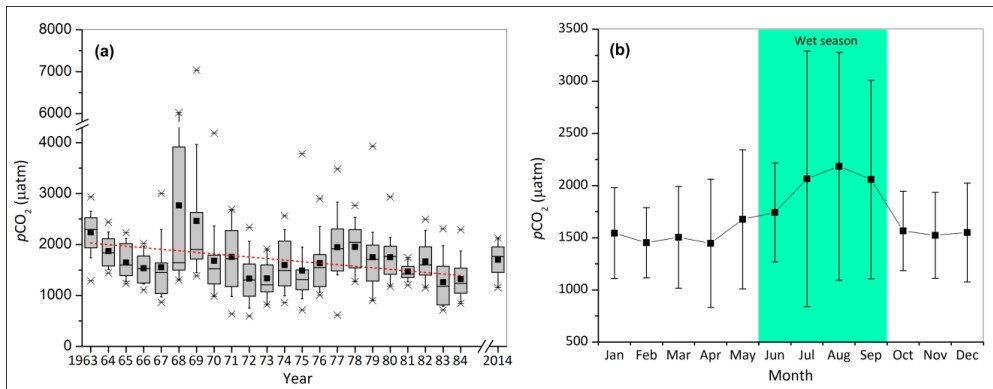

Fig. 4. Temporal variations of $p$CO$_2$ at Datong station. (a) box-and-whisker plot shows significant inter-annual changes; (b) seasonal variations. The dash line in (a) represents linear regression and the values for 2014 are derived from Liu et al. (2016). Error bars denote standard deviation.




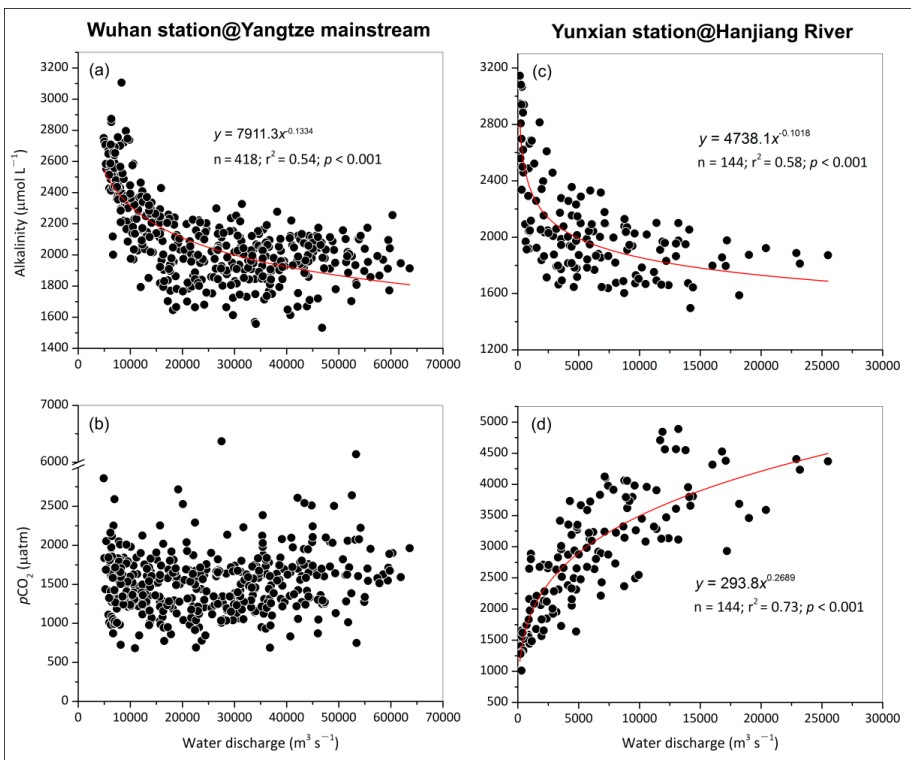

Fig. 5. Correlations between water discharge and instantaneous alkalinity and $p\mathrm{CO}_2$: the
mainstream at Wuhan station (a and b) and the Hanjiang River at Yunxian station (c and d).





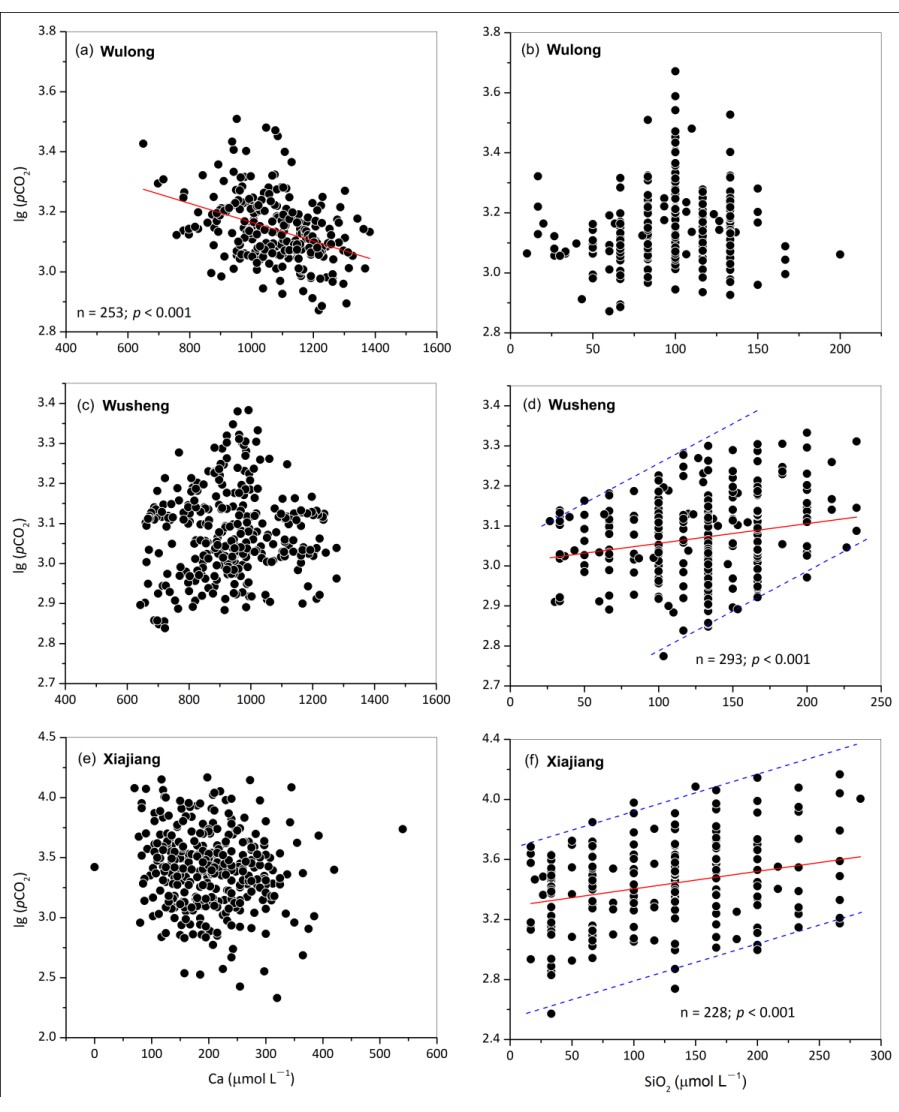

Fig. 6. Responses of $p$CO$_2$ to rock weathering products in three typical catchments with distinct
rock compositions: a–b: Wujiang River (Wulong station); b–c: Jialiangjiang River (Wusheng
station); e–f: Ganjiang River (Xiajiang station). The solid lines represent linear regression.





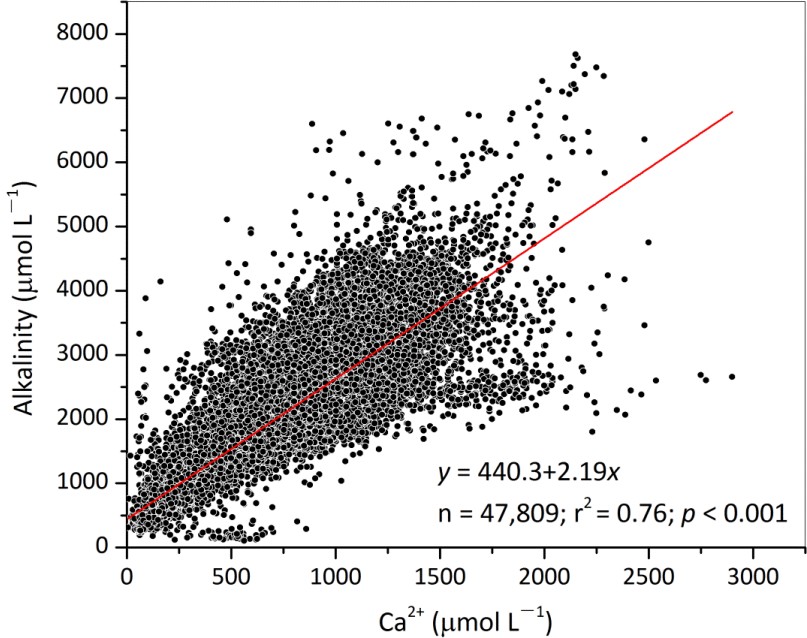

Fig. 7. Strong correlation between chemical weathering, using $Ca^{2+}$ as a proxy, and alkalinity.




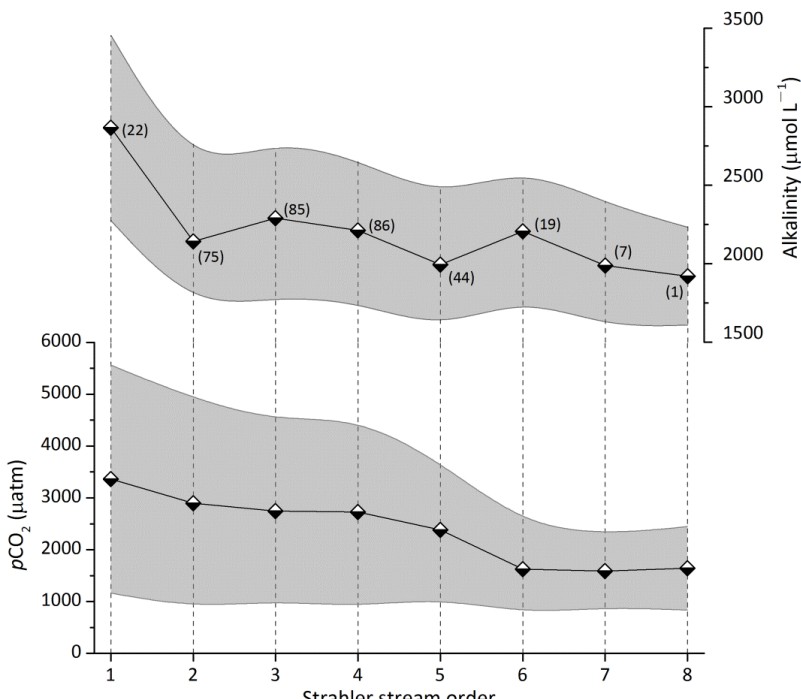

Fig. 8. Decreasing alkalinity (top) and $p$CO$_2$ (bottom) with increasing Strahler stream order. The grey shade denotes standard deviation and the numbers in parentheses represent the number of stations aggregated for each stream order.