# Peer review of "Dynamics of riverine CO2 in the Yangtze River fluvial network and their implications for carbon evasion"

_Biogeosciences, 2016_

## Referee Comment (RC1) · Anonymous Referee #1 · 13 Jan 2017

Lishan Ran and colleagues present an anaylsis of the spatial pattern in riverine pCO2 in Yangtze River basin which is representative for the time before increased anthropogenic presure by river damming operations and land-use change since the 1980's. They anaylse the correlations between Ca and Si concentrations vs pCO2 and alkalinity for different stations. They also report the long-term decrease in pCO2 and the seasonality in riverin pCO2 in the mainstem of the Yangtze river for that time. The study is, to my knowledge, novel and of interest for the scientific community. The subject would fit well within the scope of Biogeosciences. The MS is well written in most of it's parts. Methods are clearly described, figures and tables are informative. I just feel that a few more anaylsis could easily be done to make the whole study complete. That

includes a more quantitaive analysis of environmental controls of the spatial patterns in riverine pCO2, which is the main subject of this paper (see major comment #1). I suggest publication after moderate revisions.

Major comment #1 One of the main objectives of the MS is to analyze the controls of the spatial patterns in riverine pCO2. This is mainly done quite coarsely by comparing catchments that are dominated by carbonate sedimentary rocks vs. catchments dominated by other lithologies. The MS features some plots of pCO2 vs. Si and Ca concentrations or discharge (Figs 5 and 6). However, these plots are made for distinct sampling locations and what is plotted are the different samples at this location. The differences between sampling locations are then discussed considering the different environmental characteristics of the catchments. In addition, in the MS, it is mentioned that these analyses have been done for plenty of sampling locations, but only a few examples are shown. And here I do not know why these examples have been chosen and in how far they are representative for the whole data set. I would like to encourage the authors to perform a more quantitative analysis of the spatial patterns in the riverine pCO2 and its environmental controls. They could plot the average pCO2 per sampling location vs. avg. concentrations of Ca and Si per sampling location (like Humborg et al., 2010 did for Sweden) or catchment properties like climate, lithology, terrain, land use, etc., (like Lauerwald et al., 2013, did for North America). Maybe they could perform these analysis separately for different stream orders.

General comments:

Abstract

L16: Here, and throughout the MS. The unit of alkalinity is unclear. I guess you mean $\mu$eq L-1. If you want to report alkalinity as molarity, then you will have to report it as molarity of e.g. the equivalent CaCO3. But it is more common to report alkalinity in $\mu$eq L-1.

L18: 'controlled by terrestrial ecosystem'. I think you would have to be a bit more

specific, like 'C inputs from terrestrial ecosystems'.

L25: Maybe you should change 'riverine carbon' to 'riverine CO2' to be more specific and consistent with the title of the MS.

Introduction

L46: Raymond et al and Lauerwald et al. have used the same data base: GloRiCh. However, while Raymond et al. used all the calculated pCO2 values, Lauerwald et al. used only the data from 18% of the sampling locations which were selected based on a minimum number of CO2 values per sampling location.

L60-62: Is this mainly due to high soil erosion and export of particulate organic carbon? Please, clarify.

L80: Maybe add a 'the' before 'riverine carbon cycle'.

L92: 'Globally substantial' is a bit unclear to me. Maybe you could change this part of the sentence to something like 'its contribution to the global CO2 evasion from rivers is likely significant'.

L93: Maybe change 'to refine global CO2 evasion' to 'to refine estimates of global CO2 evasion'.

Methods and Materials L101-104 and Fig.1: When you talk about sedimentary rocks being mainly composed of carbonates, you should use a term like 'carbonate sedimentary rocks'. 'Carbonate' is the name of a group of minerals, but here you talk about the rocks, more precisely about the lithology. Same is true for 'Silicates'. Silicates are a group of minerals. Igneous rocks also consist mainly of silicates. And metamorphic rocks can contain silicates and/or carbonates. So, I suggest you rename the lithology to 'siliciclastic sedimentary rocks'.

L134-137: The selection of samples with a pH >6.5 itself can introduce some bias for the overall picture of spatial patterns in pCO2 and total CO2 evasion from the river

network, as some specific system might be completely excluded from the analyses. That might be inevitable, but should at some point be discussed. Here, it would be interesting how many samples have been discarded (as % of total), where the affected sampling locations are predominantly located (I see that large parts of that river system have a rather high pH, in particular where carbonate rocks are abundant), and if there are sampling locations which had to be discarded because they only have such a low pH. Note that Raymond et al., 2013 and Lauerwald et al., 2015 chose a minimum pH of 5.4. Can you argue that for so low pH values the calculation of pCO2 might already have introduced a bias in their studies?

L164: What is the conventional method? I see later that you used CO2SYS, Raymond et al., 2013 and Lauerwald et al., 2015 used PhreeqC. Would there be any systematic difference in calculated pCO2 using CO2SYS or PhreeqC? That could be answered maybe later in the discussion section.

L164-168: For what do you need the concentration of bicarbonates? Please, clarify.

Results L179-180: Maybe change to 'relatively lower' to 'relatively low'.

L182-186: Like I mentioned in the abstract, you should report your alkalinity in $\mu$eq L-1.

L193-195: If you consider the downstream decrease in pCO2 from headwaters to the lower reach of the main river, which you highlighted in the abstract, this method does not make much sense at this scale, because you ignore the stream orders of the sampled river reaches. It would make more sense if you would only interpolate the pCO2 of small headwater rivers.

L228-232: I think you are talking about lithology rather than mineralogy. See my comment in method section. L232-240: When you plot Ca or SiO2 concentrations against pCO2 per sample for distinct stations separately, then these Ca and SiO2 concentrations would represent tracers for the relative contribution of ground water inputs, which are diluted by the contributions of surface runoff (+shallow sub-surface runoff). SiO2

is likely the better tracer, because it is less reactive then Ca (which can be subject to carbonate precipitation and adsorption in the soil). Maybe you could discuss these plots a bit along those lines.

Discussion L267-276: If you exclude stream and rivers with low pH for methodological reasons, then you will systematically exclude some natural systems and have a biased estimated for the whole river network (Wallin et al. 2014, GBC). You should discuss that here as well. If you exclude form one sampling location that has pH values higher and lower 6.5, and exclude all the lower values, then you would get a biased average pCO2 at that station, in particular if you assume high pCO2 to coincide with low pH.

L296-316: Following my comment on L232-240, you could discuss the SiO2 vs pCO2 plots in Fig 6 as indication of higher CO2 concentration in ground water in Wusheng and Xiajiang catchment. For these two catchments, do you have a negative correlation between discharge and pCO2? That would be consistent with the assumption that SiO2 is a tracer for baseflow contribution vs. dilution by surface runoff. Then, these two catchments would show a different discharge-pCO2 relation than the Yunxian station. If that's the case, it would be interesting to discuss the differences. Are there riparian wetlands present upstream of Yunxian? See e.g. Teodoru et al., 2015, Biogeosciences.

L311-316: Another potential explanation is the large catchment implying a long traveling time of soil derived carbon, maybe combined with the absence of riparian wetlands around the main stem (if that is the case?). Then, direct inputs of CO2 from soil respiration and inputs of labile DOC from adjacent soils and vegetation would be relatively low. And higher inputs far upstream might have already been lost to the atmosphere.

L325-329: I do not really understand that, sorry. Could you please explain that argument in a bit more detail?

L325-334: Humborg et al., 2010 (also cited in your paper) also looked at correlations between Ca2+ and SiO2 vs pCO2. Maybe it would be good to discuss your findings with that of Humborg et al., 2010.

L345: Here, I stumbled over the term 'heterotrophic ecosystem'. Maybe you should rephrase it to 'more pronounced net-heterotrophy', or something similar.

L361: Do you mean 'km-3 yr-1', i.e. mass per year instead of area per year?

L363-365: Did soil respiration increase in response to soil erosion?

L368-370: If floodplains would be present, you would also have a positive correlation of discharge vs pCO2 in the main channel (see Mayorga et al., 2005, and Richey et al., 2002, Nature and maybe also Teodoru et al., 2015, Biogeosciences, see comment on L296-316).

L375-378: For the Amazon, Richey et al.(2002, Nature) assumes higher gas exchange velocities from large, open rivers due to wind effects. Similarly, the gas exchange velocity reported by Alin et al., 2011 for the Amazon and the Mekong basins are not generally lower for the main channel, which offers a long fetch for the wind, while smaller tributaries are more protected against wind. But these are low gradient systems (low relief). It might be different if the main control on gas exchange velocities would be the channel slope (see Raymond et al., 2012, Limnology and Oceanography). But also for larger rivers in the US it was found that gas exchange velocities were rather high and to a substantial proportion supported by wind (e.g. Beaulieu et al., 2012, JGR).

L384-385: Here you could also cite Butman and Raymond, 2011, Nat Geosc.

Conclusion L426-431: That is not the conclusion of this study but a repetition from the introduction. In the conclusion, you should simply summarize your results in order to answer the main research questions that you have worked out in the introduction.

L429-437: Here you should explain how the riverine pCO2 3 to 5 decades before today can be important for refining estimates of CO2 evasion. I guess someone estimating CO2 evasion would do it for the most recent period. Will you use these estimates in a future study to compare it to a more recent state of this system, in order to quantify the anthropogenic perturbation of the river-atmosphere CO2 fluxes due to damming and

land-use change. That would be an important outlook.

---

## Referee Comment (RC2) · Anonymous Referee #2 · 8 Mar 2017

This manuscript detailing long-term patterns of pCO2 and alkalinity in the Yangtze River is generally well-written and includes strong methodology and data. The work will be of general interest as the authors have clearly shown the relevance of this carbon component in the framework of the larger carbon cycle. My main concern with this manuscript is the overly broad interpretations of underlying processes. The very simple correlations between water chemistry and discharge with alkalinity and pCO2 certainly point towards specific processes, but the authors do not dig too deeply into these relationships. Therefore, many of the conclusions, and a fair bit of the discussion is overly speculative. I would like to see a more formal set of questions and hypotheses that could be evaluated with the data available. I believe further evaluations and rationale

will be needed to sort out the interesting pattern of stable pCO2 across a range of discharges in the mainstem (Figure 5b). The discussion of in-stream processes vs. tributary dilution and terrestrial CO2 sources is at present too speculative (lines 295-316).

———————————————————

---

## Author Comment (AC1) · 8 Mar 2017

The comment was uploaded in the form of a supplement:
http://www.biogeosciences-discuss.net/bg-2016-507/bg-2016-507-AC1-
supplement.zip
* * *

---

## Author Response (AR1)

Dear reviewer,

We thank you very much for your comments on our manuscript.
* * *
Lishan Ran and colleagues present an anaylsis of the spatial pattern in riverine pCO2 in Yangtze River basin which is representative for the time before increased anthropogenic presure by river damming operations and land-use change since the 1980's. They anaylse the correlations between Ca and Si concentrations vs pCO2 and alka-linity for different stations. They also report the long-term decrease in pCO2 and the seasonality in riverin pCO2 in the mainstem of the Yangtze river for that time. The study is, to my knowledge, novel and of interest for the scientific community. The subject would fit well within the scope of Biogeosciences. The MS is well written in most of it's parts. Methods are clearly described, figures and tables are informative. I just feel that a few more anaylsis could easily be done to make the whole study complete. That includes a more quantitaive analysis of environmental controls of the spatial patterns in riverine pCO2, which is the main subject of this paper (see major comment #1). I suggest publication after moderate revisions.

Major comment #1 One of the main objectives of the MS is to analyze the controls of the spatial patterns in riverine pCO2. This is mainly done quite coarsely by comparing catchments that are dominated by carbonate sedimentary rocks vs. catchments dominated by other lithologies. The MS features some plots of pCO2 vs. Si and Ca concentrations or discharge (Figs 5 and 6). However, these plots are made for distinct sampling locations and what is plotted are the different samples at this location. The differences between sampling locations are then discussed considering the different environmental characteristics of the catchments. In addition, in the MS, it is mentioned that these analyses have been done for plenty of sampling locations, but only a few examples are shown. And here I do not know why these examples have been chosen and in how far they are representative for the whole data set. I would like to encourage the authors to perform a more quantitative analysis of the spatial patterns in the riverine pCO2 and its environmental controls. They could plot the average pCO2 per sampling location vs. avg. concentrations of Ca and Si per sampling location (like Humborg et al., 2010 did for Sweden) or catchment properties like climate, lithology, terrain, land use, etc., (like Lauerwald et al., 2013, did for North America). Maybe they could perform these analysis separately for different stream orders.

Reply: Because the Yangtze River basin is predominantly covered with carbonate and siliciclastic sedimentary rocks (>80% of the catchment area; Figure 1), we selected the three typical catchments (Wujiang, Jialingjiang, and Ganjiang rivers) with contrasting lithologies to analyze the impact of rock weathering on DIC export and $p$CO$_2$. The Wujiang catchment is predominant with carbonate sedimentary rocks (83%) and the Ganjiang catchment with siliciclastic sedimentary rocks (65%), while the Jialingjiang catchment is in between the two rock types (Table 3). Based on your comment, we have further plotted $p$CO$_2$ against Ca$^{2+}$ and dissolved Si for the entire set of measurements using all available data pairs in the hydrological yearbooks (Figure S1 in the Supplement). Because the Yangtze River basin is characterized by significant spatial heterogeneity of lithology with different sub-catchments having different rock types, there was no discernable correlation between $p$CO$_2$ and Ca$^{2+}$ and SiO$_2$ at the whole catchment scale. This is likely because the inherent spatial heterogeneity in lithology has

obscured the signature of the $pCO_2$-$Ca^{2+}$ and $pCO_2$-Si relationships. While both positive and negative relationships existed in typical sub-catchments with predominant carbonate or siliciclastic sediment rocks, such as the carbonate-dominated Wujiang catchment and the silicate-dominated Ganjiang catchment (Figure 6), these relationships may have counteracted each other when all data points were plotted together. Similarly, we did not detect significant $pCO_2$-$Ca^{2+}$ and $pCO_2$-Si relationships for different stream orders. We have added these discussions into the manuscript and the figure into the supplement file. (lines 249-251; 367-371)

[Figure]

Relationship between (a) $pCO_2$ and $Ca^{2+}$ and (b) $pCO_2$ and dissolved $SiO_2$ in the Yangtze River basin using all available data pairs.

General comments:

Abstract
L16: Here, and throughout the MS. The unit of alkalinity is unclear. I guess you mean µeq L-1. If you want to report alkalinity as molarity, then you will have to report it as molarity of e.g. the equivalent CaCO3. But it is more common to report alkalinity in µeq L-1.
Reply: Yes, we have checked the unit of alkalinity and confirmed that it is µeq $L^{-1}$. We have revised the unit of alkalinity (µeq $L^{-1}$) throughout the manuscript. Many thanks.

L18: 'controlled by terrestrial ecosystem'. I think you would have to be a bit more specific, like 'C inputs from terrestrial ecosystems'.
Reply: The statement has been revised to 'Changes of $pCO_2$ were collectively controlled by carbon inputs from terrestrial ecosystems,'. (lines 17-19)

L25: Maybe you should change 'riverine carbon' to 'riverine CO2' to be more specific and consistent with the title of the MS.
Reply: Changed.

Introduction
L46: Raymond et al and Lauerwald et al. have used the same data base: GloRiCh. However, while Raymond et al. used all the calculated pCO2 values, Lauerwald et al. used only the data

from 18% of the sampling locations which were selected based on a minimum number of CO2 values per sampling location.

Reply: Yes, while Raymond et al. (2013) and Lauerwald et al. (2015) have used the same database, Lauerwald et al. (2015) have used much less sampling locations (17.6% of the former). We have revised this statement 'While both studies have used the same hydrochemical database (GloRiCh), it should be noted that Raymond et al. (2013) used all the calculated $pCO_2$ values whereas Lauerwald et al. (2015) used only 18% of the sampling locations.' (lines 46-48)

L60-62: Is this mainly due to high soil erosion and export of particulate organic carbon? Please, clarify.

Reply: The high contribution of Asian rivers to global carbon flux is mainly the result of their strong soil erosion and associated particulate organic carbon export. For example, Asian rivers alone accounts for 40% of the total annual sediment discharge from land to sea (Schlünz and Schneider, 2000. International Journal of Earth Sciences, 88, 599-606). We have clarified this statement in the text 'Asian rivers are significant contributors to global carbon flux as a result of high soil erosion and particulate organic carbon export, accounting for 40% of the global carbon flux from land to sea (Schlünz and Schneider, 2000; Hope et al., 1994)'. (lines 62-65)

L80: Maybe add a 'the' before 'riverine carbon cycle'.

Reply: Added.

L92: 'Globally substantial' is a bit unclear to me. Maybe you could change this part of the sentence to something like 'its contribution to the global CO2 evasion from rivers is likely significant'.

Reply: This has been changed to 'its contribution to the global $CO_2$ evasion from rivers is likely significant'. Many thanks.

L93: Maybe change 'to refine global CO2 evasion' to 'to refine estimates of global CO2 evasion'.

Reply: Changed.

Methods and Materials

L101-104 and Fig.1: When you talk about sedimentary rocks being mainly composed of carbonates, you should use a term like 'carbonate sedimentary rocks'. 'Carbonate' is the name of a group of minerals, but here you talk about the rocks, more precisely about the lithology. Same is true for 'Silicates'. Silicates are a group of minerals. Igneous rocks also consist mainly of silicates. And metamorphic rocks can contain silicates and/or carbonates. So, I suggest you rename the lithology to 'siliciclastic sedimentary rocks'.

Reply: Based on your comment, we have renamed the lithology to 'carbonate sedimentary rocks' and 'siliciclastic sedimentary rocks' throughout the text and figures.

L134-137: The selection of samples with a pH >6.5 itself can introduce some bias for the overall picture of spatial patterns in pCO2 and total CO2 evasion from the river network, as some specific system might be completely excluded from the analyses. That might be inevitable, but should at some point be discussed. Here, it would be interesting how many samples have been discarded (as % of total), where the affected sampling locations are predominantly located (I see

that large parts of that river system have a rather high pH, in particular where carbonate rocks are abundant), and if there are sampling locations which had to be discarded because they only have such a low pH. Note that Raymond et al., 2013 and Lauerwald et al., 2015 chose a minimum pH of 5.4. Can you argue that for so low pH values the calculation of pCO2 might already have introduced a bias in their studies?

Reply: To minimize the impact of noncarbonated alkalinity such as organic acid anions (Abril et al., 2015; Hunt et al., 2011), we excluded the samples with pH<6.5 from analysis. As a result, 498 measurements were discarded, accounting for ~1% of the total number of measurements (48,307), and finally 47,809 measurements were retained. The affected sampling locations were predominantly located in the lower reach (Figure 1) where strong human activities and metropolitan cities may have substantially affected its water chemistry. No sampling station was excluded solely because it had pH<6.5 samples only. Because the pH in the Yangtze River basin is rather high as a result of extensive outcrops of carbonate rocks (96% of the pH values ranged from 7.3 to 8.3), the selection criterion of pH<6.5 was used to remove the samples significantly affected by pollution. While for the global-scale $pCO_2$ estimates by Raymond et al. (2013) and Lauerwald et al. (2015), the pH variability from global inland waters is much larger, and the minimum of pH of 5.4 appears to be reasonable. For example, the natural blackwater rivers in Amazon system and SE Asian tropical catchments have much a lower pH (e.g., pH<5; Müller et al., 2015. Biogeosciences, 12, 5967-5979; Richey et al., 2002. Nature). Although a bias is inevitable, the minimum of pH=5.4 can estimate the $pCO_2$ while constraining the uncertainty as much as possible. We have further discussed the selection of samples in the revised manuscript. (lines 139-142; 149-151)

L164: What is the conventional method? I see later that you used CO2SYS, Raymond et al., 2013 and Lauerwald et al., 2015 used PhreeqC. Would there be any systematic difference in calculated pCO2 using CO2SYS or PhreeqC? That could be answered maybe later in the discussion section.

Reply: Compared with direct $pCO_2$ measurement techniques, such as the headspace equilibration technique (e.g., Müller et al., 2015. *Biogeosciences*, 20, 5967-5979; Yoon et al., 2016. *Biogeosciences*, 13, 3915-3930), here the conventional method refers to the method of using pH, alkalinity, and water temperature to calculate $pCO_2$. We have previously compared the two methods (CO2SYS and PHREEQC) in the Yellow River. The $pCO_2$ derived by CO2SYS was very close to that returned by PHREEQC program (<3% differences, Ran et al., 2015. *Biogeosciences*, 12, 921-932). While in the Yangtze River, our comparative analysis between CO2SYS and direct $pCO_2$ measurement shows a ~8% difference with the CO2SYS-based method overestimating by 8% (Liu et al., 2016. *Global Biogeochemical Cycles*. 30, 880-897). We have added this discussion into the manuscript. (lines 268-271)

L164-168: For what do you need the concentration of bicarbonates? Please, clarify.

Reply: Here we tried to emphasize the species contributing to alkalinity in the Yangtze River waters. Bicarbonates ($HCO_3^-$) dominate the alkalinity, accounting for 96%. Our recent DOC sampling analysis also suggests that the DOC in the Yangtze River is relatively low (<250 µM; Liu et al., 2016. *Global Biogeochemical Cycles*. 30, 880-897). Therefore, the impact of organic acids on the alkalinity-based $pCO_2$ calculation is predicted to be small. We have revised these statements in the manuscript for clarity. (lines 171-175)

Results L179-180: Maybe change to 'relatively lower' to 'relatively low'.
Reply: Changed.

L182-186: Like I mentioned in the abstract, you should report your alkalinity in µeq L-1.
Reply: We have revised the alkalinity in µeq $L^{-1}$ through the text.

L193-195: If you consider the downstream decrease in pCO2 from headwaters to the lower reach of the main river, which you highlighted in the abstract, this method does not make much sense at this scale, because you ignore the stream orders of the sampled river reaches. It would make more sense if you would only interpolate the pCO2 of small headwater rivers.
Reply: Based on your comment, we have selected the sampling stations located in small headwater rivers (~260 stations) and performed the Kriging interpolation to present the spatial pattern of $pCO_2$. The stations located on major tributaries and the mainstem channel were removed from the interpolation. Without the impact of mainstem stations (usually lower $pCO_2$ values), the modeled $pCO_2$ exhibited stronger spatial variability due to closer contact with the land ecosystems (449-453 µatm; Figure 3).

L228-232: I think you are talking about lithology rather than mineralogy. See my comment in method section.
Reply: Yes, here it refers to lithology, and relevant terms have been revised accordingly. Thanks.

L232-240: When you plot Ca or SiO2 concentrations against pCO2 per sample for distinct stations separately, then these Ca and SiO2 concentrations would represent tracers for the relative contribution of ground water inputs, which are diluted by the contributions of surface runoff (+shallow sub-surface runoff). SiO2 is likely the better tracer, because it is less reactive then Ca (which can be subject to carbonate precipitation and adsorption in the soil). Maybe you could discuss these plots a bit along those lines.
Reply: Based on your comment, we have discussed the correlation between $pCO_2$ and Ca and $SiO_2$ in the Discussion section. Please refer to the responses to L296-316 below. Thanks.

Discussion
L267-276: If you exclude stream and rivers with low pH for methodological reasons, then you will systematically exclude some natural systems and have a biased estimated for the whole river network (Wallin et al. 2014, GBC). You should discuss that here as well. If you exclude form one sampling location that has pH values higher and lower 6.5, and exclude all the lower values, then you would get a biased average pCO2 at that station, in particular if you assume high pCO2 to coincide with low pH.
Reply: Just as what Wallin et al. (2014) concluded, excluding the measurements with pH<6.5 values from analysis may have generated biased estimates of $pCO_2$ for the whole river network in general and some natural rivers with low pH values in particular. Considering the potential impact from human activities within the Yangtze River watershed, we removed 498 measurements from analysis, accounting for only ~1% of the total number of measurements (48,307). In addition, the used pH varied from 6.5 to 9.2 with ~96% of the pH measurements ranging from 7.3 to 8.3 (Tables 2 and S1). Thus, we concluded that the calculated $pCO_2$ is reasonable and can be used for further $CO_2$ evasion estimation. We have inserted these

justifications into the revised manuscript. In addition, we have also compiled station-based $pCO_2$ in the Supplement (Table S1). (lines 139-142; 149-151; 280-284; 286-289)

L296-316: Following my comment on L232-240, you could discuss the SiO2 vs pCO2 plots in Fig 6 as indication of higher CO2 concentration in ground water in Wusheng and Xiajiang catchment. For these two catchments, do you have a negative correlation between discharge and pCO2? That would be consistent with the assumption that SiO2 is a tracer for baseflow contribution vs. dilution by surface runoff. Then, these two catchments would show a different discharge-pCO2 relation than the Yunxian station. If that's the case, it would be interesting to discuss the differences. Are there riparian wetlands present upstream of Yunxian? See e.g. Teodoru et al., 2015, Biogeosciences.

Reply: We have further plotted the relationship between water discharge and $pCO_2$ at Wusheng (Jialingjiang catchment) and Xiajiang (Ganjiang catchment) stations (please see the figures below). Because there is only 1-year long record of discharge at Wusheng station (18 measurements in 1983), the relationship between discharge and $pCO_2$ is not as significant as that at Xiajiang station. However, the significant negative correlation between water discharge and concomitant alkalinity clearly indicates a dilution effect of surface runoff in the wet season. Just as what you expected, in the Ganjiang catchment (Xiajiang station) with dominant lithology being siliciclastic sedimentary rocks (Table 3), there is a significant negative correlation between discharge and $pCO_2$. This relationship is different from that observed at Yunxian station in Hanjiang catchment (Figure 5d), suggesting that $SiO_2$ in the Ganjiang catchment is a tracer for baseflow contribution to $pCO_2$. While in Hanjiang catchment (Yunxian station), because of the impoundment of Danjiangkou Reservoir and other smaller reservoirs, there are plenty of newly-formed floodplains and wetlands along the river and within the catchment (*Liu et al.*, 2011. Soil, Air, Water, 39, 109-115). Its positive response of $pCO_2$ to discharge indicates the importance of enhanced connectivity between river and wetlands/floodplains on river biogeochemistry, especially during wet seasons. In comparison, the deceasing $pCO_2$ at Xiajiang station with discharge is indicative of the impact of groundwater input on riverine carbon dynamics (Figs. S2 and 6f). We have added these discussion and justifications into the revised manuscript and Supplement files. (lines 324-328; 359-366)

[Figure]

Relationship between water discharge and $pCO_2$ at (a) Wusheng (Jialingjiang catchment) and (b) Xiajiang (Ganjiang catchment) stations.

L311-316: Another potential explanation is the large catchment implying a long traveling time of soil derived carbon, maybe combined with the absence of riparian wetlands around the main

stem (if that is the case?). Then, direct inputs of CO2 from soil respiration and inputs of labile DOC from adjacent soils and vegetation would be relatively low. And higher inputs far upstream might have already been lost to the atmosphere.

Reply: Many thanks for your comment. Because the mainstem channel is mainly confined to the river channel except the segments closely connected to the three lakes (see their locations in Fig. 1a), direct inputs of $CO_2$ from soil respiration and from labile DOC decomposition from adjacent soils/vegetation is relatively low. The average travel time in the Yangtze River mainstem channel is 3-5 months. Associated with much strong $CO_2$ evasion in low-order turbulent tributaries, the long travel time may have also contributed to the stable $pCO_2$ in the mainstem. We have added these potential explanations into the manuscript. (lines 328-337)

L325-329: I do not really understand that, sorry. Could you please explain that argument in a bit more detail?

Reply: We have further explained the argument: The negative correlation in Fig. 6a is contradictory to the common belief that carbonate dissolution will likely cause an elevated $pCO_2$ (Marcé et al., 2015; Teodoru et al., 2015). Given the strong correlation between $Ca^{2+}$ and alkalinity, the decreasing $pCO_2$ with increasing $Ca^{2+}$ is probably due to pH variability that may have offset the impact of weathering-induced DIC inputs in controlling $pCO_2$ (Fig. S1 in the Supplement). A slight pH increase would result in a reduced pCO2 as this calculation method is sensitive to pH fluctuations (Laruelle et al., 2013). We have also added two more references to justify the argument and two figures in the Supplement for clarity (Figure S2). (lines 347-352)

L325-334: Humborg et al., 2010 (also cited in your paper) also looked at correlations between Ca2+ and SiO2 vs pCO2. Maybe it would be good to discuss your findings with that of Humborg et al., 2010.

Reply: Humborg et al. (2010) analyzed the contributions of terrestrial respiration, chemical weathering, and aquatic respiration to $pCO_2$ in Sweden rivers and lakes. Based on your comment, we have further compared our findings with that of Humborg et al. (2010). Because weathering products are typical for groundwater input, the positive correlation between $pCO_2$ and dissolved Si suggests that riverine $pCO_2$ has a strong groundwater signature. Particularly, we analyzed the correlation between $pCO_2$ and dissolved Si in the dry season when groundwater is the major contributor to river runoff. The result shows that in the dry season $SiO_2$ can explain ~25% of the $pCO_2$ variability in the sub-catchments covered mainly with siliciclastic sediment rocks, comparable to the results by Humborg et al. (2010) in Sweden. (lines 347-352; 359-366)

L345: Here, I stumbled over the term 'heterotrophic ecosystem'. Maybe you should rephrase it to 'more pronounced net-heterotrophy', or something similar.

Reply: We have replaced the term by 'more pronounced net-heterotrophy'.

L361: Do you mean 'km-3 yr-1', i.e. mass per year instead of area per year?

Reply: The correct unit for soil erosion should be t $km^{-2}$ $yr^{-1}$ (mass per unit area per year). We have corrected the unit. Thanks.

L363-365: Did soil respiration increase in response to soil erosion?

Reply: Because of the high soil erosion rate (2167 t/km$^2$/yr), huge amounts of organic carbon is discharged into the river network. The availability of organic carbon during fluvial delivery

would enhance decomposition of organic carbon and the production of $CO_2$. We have revised this justification in the revised manuscript 'decomposition of the terrestrial-origin organic carbon has resulted in the $CO_2$ excess in the headwater streams (Li et al., 2012)'. (lines 400-401)

L368-370: If floodplains would be present, you would also have a positive correlation of discharge vs pCO2 in the main channel (see Mayorga et al., 2005, and Richey et al., 2002, Nature and maybe also Teodoru et al., 2015, Biogeosciences, see comment on L296-316).
Reply: Yes, presence of floodplains and the enhanced connectivity between river and floodplains will cause a positive correlation between $pCO_2$ and water discharge as observed at Yunxian station in Hanjiang catchment. Because of the construction of Danjiangkou Reservoir (storage: 17.5 km$^3$) and other smaller reservoirs, there are widespread presence of wetlands and floodplains along the river channel. The observed positive relationship at Yunxian (Fig. 5d) reflected the importance of wetland/floodplains in affecting $pCO_2$. This discussion has been added into the manuscript. Thanks. (lines 324-328)

L375-378: For the Amazon, Richey et al. (2002, Nature) assumes higher gas exchange velocities from large, open rivers due to wind effects. Similarly, the gas exchange velocity reported by Alin et al., 2011 for the Amazon and the Mekong basins are not generally lower for the main channel, which offers a long fetch for the wind, while smaller tributaries are more protected against wind. But these are low gradient systems (low relief). It might be different if the main control on gas exchange velocities would be the channel slope (see Raymond et al., 2012, Limnology and Oceanography). But also for larger rivers in the US it was found that gas exchange velocities were rather high and to a substantial proportion supported by wind (e.g. Beaulieu et al., 2012, JGR).
Reply: High gas transfer velocities were observed in large, open rivers in Amazon and Mekong river catchments (Richey et al., 2002; Alin et al., 2011). These studies are performed on the mainstem channel and primary tributaries where flow velocity is generally lower than the lower-order upstream rivers and streams due to gentler channel slopes. Therefore, wind speed appears to be the predominant factor affecting gas transfer as in lake and reservoir settings. In comparison, the gas transfer velocity in lower-order streams is more controlled by channel slope and thus flow velocity (Borges et al., 2015. Nat Geoscience; Butman and Raymond, 2011. Nature Geoscience; Raymond et al., 2012. Limnology and Oceanography). The Yangtze River basin typically has a low wind speed (<2 m/s; see the figure below, adapted from Gong et al., 2006. Journal of Hydrology, 329, 620-629), lower than the measurements obtained by Alin et al. (2011), while its flow velocity varying from 0.3 to 2.3 m/s with low values mostly observed in the mainstem channel is significantly higher than that reported by Beaulieu et al. (2012) (i.e., 0-0.8 m/s). Therefore, the gas transfer velocity on the mainstem is likely lower than in the steep lower-order streams, and the $CO_2$ efflux from the mainstem water surface is also likely lower.

[Figure]

Mean daily variations of wind speed in the upper (U), middle (M) and lower reaches (L) of the Yangtze River basin. Adapted from Gong et al. (2006).

L384-385: Here you could also cite Butman and Raymond, 2011, Nat Geosc.
Reply: This reference has been added into the text to enhance the justification.

Conclusion
 L426-431: That is not the conclusion of this study but a repetition from the introduction. In the conclusion, you should simply summarize your results in order to answer the main research questions that you have worked out in the introduction.
Reply: Based on your comment, we have removed the repetition sentences and summarized the results from this study.

L429-437: Here you should explain how the riverine pCO2 3 to 5 decades before today can be important for refining estimates of CO2 evasion. I guess someone estimating CO2 evasion would do it for the most recent period. Will you use these estimates in a future study to compare it to a more recent state of this system, in order to quantify the anthropogenic perturbation of the river-atmosphere CO2 fluxes due to damming and land-use change. That would be an important outlook.
Reply: Extensive and intensive human disturbances, mainly damming and land-use change, have occurred within the catchment in the most recent decades (since the 1990s). Our next step is to estimate $CO_2$ emissions across the water-air interface from this river network by using the computed $p$$CO_2$ in this study. Comparing the $CO_2$ evasion before large-scale human impacts with recent evasion estimates will allow us to quantify anthropogenic perturbations of the river-atmosphere $CO_2$ fluxes. Although a catchment-scale $CO_2$ evasion estimate remains unknown, there are a few studies on the mainstem or on sub-catchments (e.g., Li et al., 2012. Journal of Hydrology, 466-467, 141-150; Liu et al., 2016. Global Biogeochemical Cycles, 30, 880-897). We have revised the statement to 'Given the extensive and intensive human disturbances within the watershed since the 1990s, special attention must be paid to the resulting changes to riverine $p$$CO_2$ and $CO_2$ evasion. A comparative analysis involving $CO_2$ evasion before large-scale human impacts and recent evasion estimates (e.g., Li et al., 2012; Liu et al., 2016) will be able to examine the anthropogenic perturbations of the river-atmosphere $CO_2$ fluxes due to damming and land-use change'. Many thanks for your comments. (lines 487-492)

Dear reviewer,

We thank you very much for your comments on our manuscript.
* * *
This manuscript detailing long-term patterns of pCO2 and alkalinity in the Yangtze River is generally well-written and includes strong methodology and data. The work will be of general interest as the authors have clearly shown the relevance of this carbon component in the framework of the larger carbon cycle. My main concern with this manuscript is the overly broad interpretations of underlying processes. The very simple correlations between water chemistry and discharge with alkalinity and pCO2 certainly point towards specific processes, but the authors do not dig too deeply into these relationships. Therefore, many of the conclusions, and a fair bit of the discussion is overly speculative. I would like to see a more formal set of questions and hypotheses that could be evaluated with the data available. I believe further evaluations and rationale will be needed to sort out the interesting pattern of stable pCO2 across a range of discharges in the mainstem (Figure 5b). The discussion of in-stream processes vs. tributary dilution and terrestrial CO2 sources is at present too speculative (lines 295-316).

Reply: Based on your comments, we have further analyzed the underlying processes controlling riverine $pCO_2$ changes in the Yangtze River watershed. In addition to the discussion based on the three typical sub-catchments with contrasting lithologic features (i.e., Wujiang, Jialingjiang, and Ganjiang catchments), we have also examined the relationships between $pCO_2$ and representative water chemistry variables (e.g., $Ca^{2+}$ and $SiO_2$) at the entire watershed scale (Fig. S1 in the Supplement). The indiscernible $pCO_2$-$Ca^{2+}$ and $pCO_2$-$SiO_2$ relationship for the entire watershed may be attributed to the spatial heterogeneity in lithology that has obscured the signature. While both positive and negative relationships existed in sub-catchments with predominant carbonate or siliciclastic sediment rocks (Figs. 6 and S3), these relationships may have counteracted each other when all data points were plotted together. For the $pCO_2$ changes in the mainstem channel (Fig. 5b), it is likely because the increased dissolved $CO_2$ inputs by soil organic matter decomposition from one region has been counteracted by low $pCO_2$ waters derived from other regions. This is highly possible given its heterogeneous catchment settings in terms of vegetation cover, soil type, and rainfall intensity. Furthermore, the large catchment implies a long travel time of land-derived organic carbon during fluvial delivery (3-5 months). Coupled with limited floodplains along the mainstem channel (please refer to lines 407-407), direct inputs of $CO_2$ from soil respiration would be relatively low whereas strong $CO_2$ evasion in lower-order turbulent tributaries might have already exhausted dissolved $CO_2$. Moreover, the mainstem rivers are generally characterized by comparatively low gas transfer velocities due to weakened turbulence and mixing with benthic substrates (Butman and Raymond, 2011; Borges et al., 2015), which can effectively inhibit $CO_2$ degassing and therefore maintain the balance. An example is the Yangtze estuary that presents considerably low $CO_2$ evasion fluxes of 16-34 mol m$^{-2}$ yr$^{-1}$, despite its significantly higher riverine $pCO_2$ than the overlying atmosphere (Zhai et al., 2007. Marine Chemistry,107, 342-356). Thus, its $pCO_2$ dynamics appeared to be independent of hydrograph. Clearly, this stable $pCO_2$ regardless of water discharge changes is different from that in tributaries. (please refer to lines 328-337)

Moreover, we further conducted a comparative analysis regarding the differences in $pCO_2$ among sub-catchments by relating to their hydrological connectivity, flow regime and $CO_2$ sources,

including in-stream processing, terrestrial $CO_2$ sources, and dilution in wet seasons. In addition to the Hanjiang (Figs. 5c and 5d), we plotted the relationship between water discharge and $pCO_2$ at Wusheng (Jialingjiang catchment) and Xiajiang (Ganjiang catchment) stations (please see Figure A below) to analyze the underlying processes controlling $pCO_2$. Because there is only 1-year long record of discharge at Wusheng station (18 measurements in 1983), the relationship between discharge and $pCO_2$ is not as significant as that at Xiajiang station. However, the significant negative correlation between water discharge and concomitant alkalinity clearly indicates a dilution effect of surface runoff in the wet season. For the Ganjiang catchment (Xiajiang station) with dominant lithology being siliciclastic sedimentary rocks (Table 3 in the manuscript), there is a significant negative correlation between discharge and $pCO_2$ (please refer to Figure A_b below, also included in the Supplement (Fig. S3)). This relationship is different from that observed at Yunxian station in Hanjiang catchment (Fig. 5d), suggesting that $SiO_2$ in the Ganjiang catchment is a tracer for baseflow contribution to $pCO_2$. The deceasing $pCO_2$ at Xiajiang station with discharge is indicative of the impact of groundwater input on riverine carbon dynamics (Figs. S2 and 6f). We have added these discussion and justifications into the revised manuscript and Supplement files. (lines 324-328; 359-366)

[Figure]

Figure A: Relationship between $pCO_2$ and water discharge at (a) Wusheng (Jialingjiang catchment) and (b) Xiajiang (Ganjiang catchment) stations.

With respect to the impact of hydrologic connectivity, presence of wetlands and floodplains affects river biogeochemistry (Teodoru et al., 2015. Biogeosciences, 12, 2431-2453). For example, because of the impoundment of Danjiangkou Reservoir and other smaller reservoirs, there are plenty of newly-formed floodplains and wetlands along the river and within the Hanjiang catchment (*Liu et al*., 2011. Soil, Air, Water, 39, 109-115). The observed positive response of $pCO_2$ to discharge at Yunxian station indicates the importance of enhanced connectivity between river and wetlands/floodplains on river biogeochemistry, especially during wet seasons (Fig. 5d). That is, the enhanced connectivity between river and wetlands/floodplains along aquatic continuum, especially during wet seasons, has maintained the high $pCO_2$ levels (Fig. 5d), as has been observed by Abril et al. (2014. Nature, 505, 395-398) in the Amazon River. A comparative analysis between Hanjiang and Ganjiang catchments suggests the differences in underlying processes influencing riverine $pCO_2$. Particularly, in dry seasons with groundwater dominating the runoff, $SiO_2$ serves as a good tracer of groundwater inputs and can explain ~25% of the $pCO_2$ variability in sub-catchments covered mainly with siliciclastic sediment rocks, such

as the Ganjiang catchment (see Figure B below). This is comparable to the results by Humborg et al. (2010) in Sweden. (lines 359-366).

[Figure]

Figure B: Correlation between water discharge and dissolved $SiO_2$ at Xiajiang station (Ganjiang catchment) with high $SiO_2$ concentrations (e.g., >200 µmol $L^{-1}$) primarily observed in low flow periods (dry seasons).

Overall, we have further investigated the underlying processes affecting riverine $pCO_2$ within the Yangtze River basin by more systematically exploring the relationships between $pCO_2$ and various environmental variables. These discussions have been added into the revised manuscript or supplement, and related references have also been added to justify our arguments. In addition, to make the computed $pCO_2$ data be accessible to the public for global-scale $CO_2$ evasion estimate, we have also summarized the 339 station-based $pCO_2$ in the Supplement (Table S1). Major changes and additions have been highlighted in the revised version of the manuscript. Thanks again for your constructive comments.

[revised manuscript text omitted]